# Carbonate-silicate cycle predictions of Earth-like planetary climates and testing the habitable zone concept

Owen R. Lehmer [1,2,3✉], David C. Catling [2,3] & Joshua Krissansen-Totton[3,4]

In the conventional habitable zone (HZ) concept, a $CO_2$-$H_2O$ greenhouse maintains surface liquid water. Through the water-mediated carbonate-silicate weathering cycle, atmospheric $CO_2$ partial pressure ($pCO_2$) responds to changes in surface temperature, stabilizing the climate over geologic timescales. We show that this weathering feedback ought to produce a log-linear relationship between $pCO_2$ and incident flux on Earth-like planets in the HZ. However, this trend has scatter because geophysical and physicochemical parameters can vary, such as land area for weathering and $CO_2$ outgassing fluxes. Using a coupled climate and carbonate-silicate weathering model, we quantify the likely scatter in $pCO_2$ with orbital distance throughout the HZ. From this dispersion, we predict a two-dimensional relationship between incident flux and $pCO_2$ in the HZ and show that it could be detected from at least 83 ($2\sigma$) Earth-like exoplanet observations. If fewer Earth-like exoplanets are observed, testing the HZ hypothesis from this relationship could be difficult.

[1] MS 239-4, Space Science Division, NASA Ames Research Center, Moffett Field, CA 94035, USA. [2] Department of Earth and Space Sciences/Astrobiology Program, University of Washington, Box 351310, Seattle, WA 98195, USA. [3] Virtual Planetary Laboratory at the University of Washington, Seattle, WA 98195, USA. [4] Department of Astronomy and Astrophysics, MS UCO/Lick Observatory, 1156 High Street, Santa Cruz, CA 95064, USA. ✉email: owen.r.lehmer@nasa.gov

N ewton first alluded to the concept of a stellar habitable zone (HZ) in his 1687 Principia[1] by noting that Earth's liquid water would vaporize or freeze at the orbits of Mercury and Saturn, respectively[2]. Later, Whewell noted that "the Earth's orbit is in the temperate zone of the Solar System"[3]. Since then, the definition of the stellar HZ has been refined, reaching its modern incarnation based on climate models[4,5].

Current HZ calculations[6] find that around a Sun-like star, an Earth-like planet could remain habitable between 0.97 and 1.70 AU. The inner edge of the HZ is set by loss of surface water and the outer edge is set by the maximum greenhouse of a $CO_2$ atmosphere where extensive $CO_2$ condensation and increased Rayleigh scattering prevent any further greenhouse warming from $CO_2$ (refs. [6,7]). This definition of the HZ only considers $H_2O$ and $CO_2$ as greenhouse gases, so Earth-like planets warmed by other greenhouse gases (e.g., $H_2$ or $CH_4$) could remain habitable at bigger orbital distances[5,8,9]. However, $CH_4$-rich atmospheres in the HZ may not be possible without life to generate substantial $CH_4$ (refs. [10,11]). In addition, more complex climate models have shown the HZ might extend to smaller orbital distances, perhaps interior to Venus' orbit, with appropriate planetary conditions[12–15].

Residing within the HZ does not guarantee habitable surface conditions. Crucially, greenhouse gas abundances (and planetary albedo) must conspire to produce clement surface conditions. For example, by most estimates, Mars resides within the Sun's HZ but is not habitable because there is insufficient greenhouse warming from $CO_2$, in part because of the lack of volcanic outgassing of $CO_2$. Thus, considering the planetary processes that control atmospheric $CO_2$ abundances on Earth-like planets in the HZ is necessary to constrain planetary habitability.

The prevailing hypothesis is that $CO_2$ levels are controlled by a weathering thermostat[16]. This can explain why Earth has maintained a clement surface throughout its history despite the ~30% brightening of the Sun over the past ~4.5 Gyr[17–21]. The changing luminosity of the Sun with time is similar to moving a planet through the HZ, and so the same $CO_2$ weathering process responsible for maintaining habitability on the Earth through time, the carbonate–silicate weathering cycle, may similarly stabilize planetary climates within the HZ.

In the carbonate–silicate cycle, atmospheric $CO_2$ dissolves in water and weathers silicates on both the continents and seafloor, which releases cations and anions[16,22–27]. On the continents, the weathering products, including dissolved $SiO_2$, $HCO_3^-$, and $Ca^{++}$, wash into the oceans where the $HCO_3^-$ combine with cations like $Ca^{++}$ to create $CaCO_3$, which precipitates out of solution. The net process converts atmospheric $CO_2$ into marine carbonate minerals (i.e., $CaCO_3$). Also, seafloor weathering occurs when seawater releases $Ca^{++}$ ions from the seafloor basalt and $CaCO_3$ precipitates in pores and veins. Subsequently, the carbonates within sediments and altered seafloor can be subducted.

Carbon returns to the atmosphere from outgassing. If $CO_2$ outgassing increases above the steady-state outgassing rate, a planet's surface temperature rises. This leads to increased rainfall and continental weathering as well as potentially warmer deep-sea temperatures and more seafloor weathering[21,24,28]. Increased weathering draws down atmospheric $CO_2$ and stabilizes the climate over ~$10^6$-year timescales on habitable, Earth-like planets[29].

One- and three-dimensional (1D, 3D) climate calculations of HZ limits[4,6,14] assume that a carbonate–silicate weathering cycle is functioning but do not explicitly include it. The assumed presence of the carbonate–silicate cycle would predict that atmospheric $CO_2$ of Earth-like planets increases with orbital distance in the HZ[4,6,29]. In particular, future telescopic observations, e.g., NASA's Habitable Exoplanet Imaging Mission (HabEx)[30] and Large Ultraviolet Optical Infrared Surveyor

(LUVOIR)[31], could search for the $CO_2$ trend to test the HZ hypothesis[32–34]. Previous studies[29,35] have suggested the carbonate–silicate weathering cycle could alter predictions of $pCO_2$ in the HZ, but it is important to know the exact relationship we are looking for. Also, while an increase of $pCO_2$ with orbital distance in the HZ may be true if all Earth-like exoplanets have the exact same properties as the modern Earth, the trend becomes less certain if HZ planetary characteristics deviate from those of the modern Earth. There could be considerable variability in atmospheric $CO_2$ throughout the HZ, perhaps even enough to obscure a monotonic trend with orbital distance.

Here, we show that uncertain physicochemical and geophysical parameters in the carbonate–silicate weathering cycle[26] cause scatter in $pCO_2$ with orbital distance. We then demonstrate that future telescopes must observe at least 83 ($2\sigma$) HZ planetary atmospheres to confidently detect our predicted relationship between atmospheric $CO_2$ and orbital distance, and confirm the HZ hypothesis.

## Results

**Stable $pCO_2$ abundances from our numerical model**. We use a coupled climate and carbonate–silicate weathering model (see Methods, subsection "Numerical carbonate–silicate cycle modeling") to explore $pCO_2$ on Earth-like planets in the HZ. The model considers numerous planetary properties, listed in Table 1, and their effect on the carbonate–silicate weathering cycle to calculate a planet's steady-state $pCO_2$ and surface temperature. If the globally averaged, steady-state surface temperature is below 248 K, we assume the planet is completely frozen and uninhabitable at the surface, as shown by 3D climate models[36]. Similarly, we assume planets are uninhabitable beyond 355 K, above which surface water would be rapidly lost to space[37] (see Methods, subsection "Numerical carbonate–silicate cycle modeling" for additional details on these assumed temperature constraints).

We randomly generated 1050 habitable, stable, Earth-like exoplanet climates using uniform distributions of the model parameters in Table 1. A total of 1200 random, initial parameter combinations were considered but we eliminated those that resulted in planets that froze completely or were too hot to retain their surface oceans. As colored dots, Fig. 1 shows habitable, steady-state solutions.

Our model predicts that atmospheric $CO_2$ abundances should broadly increase and narrow in their spread with orbital distance in the HZ (Fig. 1), consistent with other models of $CO_2$ in the HZ[29,38]. As justified next in section "Habitable zone climate theory revisited", the scatter is about a nominal linear trend between incident flux, $S$, and log($pCO_2$), which is different from a non-linear trend in models that assume a constant surface temperature in the HZ from negative feedbacks[32,34] but do not actually model the carbonate–silicate feedbacks. If future missions are to test the HZ concept by searching for a trend between incident flux, $S$, and $pCO_2$ (refs. [32–34]), they could search for the fundamental $S$–$pCO_2$ relationship shown in Fig. 1.

Below, we show that a log-linear relationship between $pCO_2$ and $S$ may be the default in the HZ if Earth-like carbonate–silicate weathering is ubiquitous on habitable planets. In fact, the trend is elucidated by combining climate theory with carbonate–silicate cycle theory in what follows.

**Habitable zone climate theory revisited**. A conventional assumption is that the carbonate–silicate weathering cycle will approximately maintain a stable, temperate surface temperature for an Earth-like planet moved about in the HZ[6,7,39] or even a constant temperature[32,34]. Thus, if we moved the modern Earth outward in the HZ, the smaller incident flux would initially cause

**Table 1 Parameter ranges for our numerical model.**

| Parameter | Parameter description | Range | Scaling | Units |
|---|---|---|---|---|
| $F_{out}^{mod}$ | Modern $CO_2$ outgassing flux | 6–10 | | Tmol C yr$^{-1}$ |
| $n$ | Carbonate precipitation coefficient | 1–2.5 | $\propto \left[CO_3^{2-}\right]^n$ | |
| $x$ | Modern seafloor dissolution relative to precipitation | 0.5–1.5 | $\propto x F_{out}^{mod}$ | |
| $T_e$ | E-folding temperature factor for continental weathering | 10–40 | | K |
| $\alpha$ | Power law exponent for $CO_2$ dependence of continental silicate weathering | 0.1–0.5 | $\propto (pCO_2)^\alpha$ | |
| $\xi$ | Power law exponent for $CO_2$ dependence of continental carbonate weathering | 0.1–0.5 | $\propto (pCO_2)^\xi$ | |
| $f_{land}$ | Land fraction compared to modern Earth | 0–1 | | |
| $S_{thick}$ | Ocean sediment thickness relative to modern Earth | 0.2–1 | | |
| $F_{carb}^{mod}$ | Modern continental carbonate weathering | 7–14 | | Tmol C yr$^{-1}$ |
| $f_{bio}$ | Biological weathering compared to modern Earth | 0–1 | | |
| $a_{grad}$ | Surface to deep ocean temperature gradient scaling | 0.8–1.4 | $\propto a_{grad} T_s$ | |
| $\gamma$ | Power law exponent for pH dependence of seafloor dissolution | 0–0.5 | $\propto \left(\left[H^+\right]\right)^\gamma$ | |
| $\beta$ | Power law exponent for seafloor spreading rate | 0–0.2 | $\propto Q^\beta$ | |
| $m$ | Exponent for outgassing dependence on crustal production | 1–2 | $\propto Q^m$ | |
| $E_{bas}$ | Seafloor dissolution activation energy | 60–100 | | kJ mol$^{-1}$ |
| $n_{out}$ | Exponent for internal heat with time | 0–0.73 | see Eq. (10) | |
| $\tau$ | Planet age (see Eq. (10))* | 0–10 | | Gyr |
| $S$ | Incident flux relative to modern Earth* | 0.35–1.05 | | |

Parameters are dimensionless unless otherwise described. The fourth column shows how scaling parameters impact the model, where $T_s$ is the surface temperature in K and $Q$ is the internal heat of the planet relative to the modern Earth (see Eq. (10) for $Q$). Unless otherwise noted, each parameter range is justified in the original model derivation for the Earth through time[21].
*The justification for this parameter is given in the Methods, subsection "Numerical carbonate–silicate cycle modeling".

the planet to cool. The cooler temperature would lower the $CO_2$ weathering rate causing $CO_2$ to accumulate in the atmosphere until the temperature returned to its nominal value of 289 K. Figure 2 shows this scenario with the dotted blue 289 K contour, which gives the pCO$_2$ value required to maintain a 289-K surface temperature for the modern Earth as it moves about the HZ. The line was calculated from a radiative-convective climate model described in the Methods below, subsection "Habitable zone 1D climate model" (see Eq. (8)).

The constant, 289 K surface temperature contour in Fig. 2 is a non-linear relationship between incident flux, $S$, and log(pCO$_2$) but it does not consider the temperature and pCO$_2$ feedbacks inherent to the carbonate–silicate weathering cycle. We demonstrate that if these feedbacks are taken into account, surface temperature declines with orbital distance, as mentioned in previous work[29], and the relationship between $S$ and log(pCO$_2$) is actually approximately linear for Earth-like planets in the HZ.

If Bond albedo is fixed, the surface temperature, $T_s$, for an Earth-like planet in steady state varies approximately linearly with incident flux, $S$[5,40,41]. This linear relationship between $T_s$ and $S$ arises from energy balance and from water vapor feedback and can be expressed as

$$F_{SOL} = F_{OLR} = \left(\frac{1 - A_B}{4}\right)S = a + bT_s, \qquad (1)$$

where $F_{SOL}$ is the incoming solar radiation flux, $F_{OLR}$ the outgoing long-wavelength radiation flux, $A_B$ the Bond albedo, and $a$ and $b$ are empirical constants. From satellite measurements of the modern Earth and radiative calculations, for $T_s$ in K, the empirical constants in Eq. (1) are approximately $a = -370$ W m$^{-2}$ and $b = 2.2$ W m$^{-2}$ K$^{-1}$ (ref. [41]).

Solving for $T_s$ in Eq. (1), the surface temperature is given by

$$T_s = \left(\frac{1 - A_B}{4b}\right)S - \frac{a}{b}. \qquad (2)$$

Under the conventional assumption that the HZ is regulated by a $CO_2$–$H_2O$ greenhouse effect where $H_2O$ concentrations respond to changes in pCO$_2$, the temperature offset in Eq. (2), $-a/b$, is a

function of pCO$_2$. Thus, surface temperature, as a function of $S$ and pCO$_2$, is given by

$$T_s(S, pCO_2) = \left(\frac{1 - A_B}{4b}\right)S + f(pCO_2), \qquad (3)$$

where $f(pCO_2)$ is a function that depends on pCO$_2$. For the modern Earth at 1 AU, $f(pCO_2) = -a/b$. For pCO$_2 \leq 0.1$ bar, the $CO_2$ greenhouse effect is logarithmic in pCO$_2$, i.e., $f(pCO_2) \propto \ln(pCO_2)$[42,43]. Above ~0.1 bar, weaker $CO_2$ absorption features become important and $f(pCO_2)$ deviates from $\propto \ln(pCO_2)$[43,44].

As pCO$_2$ increases for an Earth-like planet moved outward in the HZ, the surface temperature will follow Eq. (3) while the rate at which $CO_2$ is removed from the atmosphere will adjust according to the carbonate–silicate weathering feedback. Quantitatively, the pCO$_2$- and $T_s$-dependent flux of $CO_2$ removal due to the continental weathering flux, $F_w$ (in mol $CO_2$ per unit time) is described by

$$F_w = \rho \left(\frac{pCO_2}{pCO_2^{mod}}\right)^\alpha \exp\left(\frac{T_s(S, pCO_2) - T_s^{mod}}{T_e}\right), \qquad (4)$$

where $\rho$ is a constant determined by the continental weathering properties of the modern Earth, $\alpha$ a dimensionless constant between 0.1 and 0.5 and regulates the pCO$_2$ dependence of continental silicate weathering, $T_e$ a constant between 10 K and 40 K and represents the e-folding temperature dependence of continental weathering. The range for $T_e$ has been empirically constrained for the surface temperatures relevant to habitable, Earth-like planets from lab measurements and Phanerozoic geologic constraints[16,26]. Finally, pCO$_2^{mod} = 288 \times 10^{-6}$ bar and $T_s^{mod} = 289$ K are the modern Earth's preindustrial pCO$_2$ and surface temperature, respectively[21].

The range for $\alpha$ on the Earth was determined empirically from geologic constraints over the past 100 Myr[26]. We assume that this derived range for $\alpha$ applies to the Earth through time[21,45,46] and the Earth-like exoplanets modeled here that have a carbonate–silicate cycle. However, better proxy data for the ancient Earth or observing the carbonate–silicate cycle on

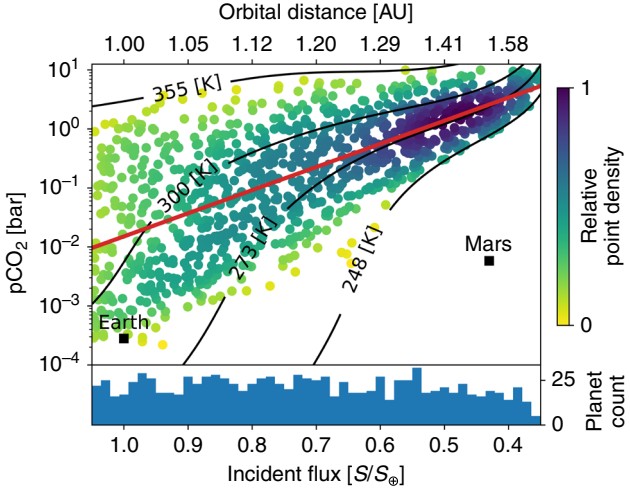

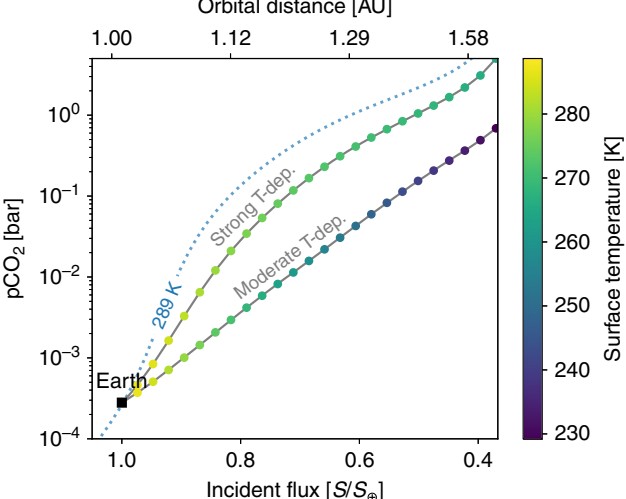

**Fig. 1 The expected distribution of stable, Earth-like exoplanet climates from our habitable zone weathering model.** The horizontal axis shows incident flux, $S$, normalized to the solar constant ($S_\oplus$) and the corresponding orbital distance in Astronomical Units (AU) above the plot. The vertical axis shows the atmospheric $CO_2$ partial pressure (p$CO_2$) in bar. Each point represents a climate in steady state. The black labeled contours show the mean global surface temperature for the given p$CO_2$ and incident flux. The white region below the 248 K contour is where our model assumption of a liquid ocean is no longer plausible so no planets are shown in that region. Above the 355 K contour, Earth-like planets are too hot to retain their liquid oceans for billions of years. Similar to the frozen planets, such hot planets are not considered habitable. Modern Earth and Mars are shown by black squares. The blue histogram at the bottom of the figure shows the number of stable planets in each incident flux bin. The color of each simulated planet shows the relative point density in the plot at that location. The color was calculated using a kernel-density estimate with Gaussian kernels and rescaled from 0 to 1. A color value of 0 represents the lowest relative point density, 1 the highest. The log-linear line of best fit between p$CO_2$ and $S$ is shown in red. The slope of the red, best fit line is $3.92 \pm 0.24$ (95%) with units $-\log_{10}$(p$CO_2$ [bar])/($S/S_\oplus$). Our model predicts that atmospheric $CO_2$ should increase with orbital distance in the HZ.

habitable exoplanets[32,34] may be necessary to understand if the assumed range for $\alpha$ applies more generally to habitable planets.

In Eq. (4), we assume seafloor weathering is negligible, which is a reasonable approximation for the modern Earth[21], and illustrative for our purposes of deriving a simple, analytic relationship between $S$ and p$CO_2$. Here, we seek to predict the behavior of the modern Earth in the HZ to gain intuitive understanding, whereas in our numerical model we consider a broad range of properties for Earth-like planets on which seafloor weathering may be important.

The modern Earth, and all Earth-like planets considered in this work, are assumed to be in steady state, in which the flux of $CO_2$ from volcanic outgassing is equal to the rate of $CO_2$ removal from weathering, $F_w$. If we assume a HZ planet with $CO_2$ outgassing the same as the modern Earth's, $F_w$ remains constant despite changes in $S$ and p$CO_2$. Setting $T_s(S, pCO_2) = T_s^{mod}$ and $pCO_2 = pCO_2^{mod}$ for the modern Earth, from Eq. (4), we see that $F_w = \rho$ and

$$1 = \left(\frac{pCO_2}{pCO_2^{mod}}\right)^\alpha \exp\left(\frac{T_s(S, pCO_2) - T_s^{mod}}{T_e}\right). \quad (5)$$

Equation (5) must hold for a modern Earth within the HZ. If it did not, $F_w$ would not balance $CO_2$ outgassing, which would result in either complete $CO_2$ removal, or $CO_2$ accumulation without bound.

**Fig. 2 The relationship between incident flux and atmospheric $CO_2$ for Earth-like planets regulated by a carbonate–silicate weathering cycle.** The horizontal axis shows incident flux, $S$, normalized to the solar constant ($S_\oplus$) and the corresponding orbital distance in Astronomical Units (AU) above the plot. The vertical axis shows the atmospheric $CO_2$ partial pressure (p$CO_2$) in bar. The dotted blue curve labeled 289 K shows the p$CO_2$ value required to maintain a 289 K surface temperature for the given incident flux, $S$. The conventional assumption of $CO_2$ in the HZ stipulates that p$CO_2$ will adjust to maintain a temperate or even constant surface temperature. Under this assumption, moving the modern Earth (labeled black square) outward in the HZ would have the planet approximately follow the dotted blue 289 K contour. The colored points and gray curves show the modern Earth moving outward in the HZ with a functioning carbonate–silicate weathering cycle, calculated from Eq. (6). We consider two temperature and p$CO_2$ dependencies for continental weathering in this plot. The strong temperature dependence contour (labeled Strong T-dep.), uses a temperature and p$CO_2$-dependent weathering factor of $\alpha T_e = 2.3$, which implies a strong temperature feedback on continental weathering compared to the p$CO_2$ feedback (see Eq. (7)). The moderate temperature dependence contour (labeled Moderate T-dep.), uses a temperature and p$CO_2$-dependent weathering factor of $\alpha T_e = 7.5$. These two values for $\alpha T_e$ result in two different paths the Earth can take as it moves outward in the HZ. The planet color shows the mean surface temperature. Log-linear fits to the colored points of the Strong T-dep. and Moderate T-dep. contours have $r^2$ values of 0.959 and 0.999, respectively. Thus, even for a strong temperature dependence of continental weathering, our coupled climate and weathering model predicts an approximately log-linear relationship between incident flux and p$CO_2$ on Earth-like planets in the HZ.

Solving for $T_s(S, pCO_2)$ in Eq. (5), we find

$$T_s(S, pCO_2) = T_s^{mod} - \alpha T_e \ln\left(\frac{pCO_2}{pCO_2^{mod}}\right). \quad (6)$$

Equating Eq. (6) to Eq. (3) and rearranging gives

$$\left(\frac{1 - A_B}{4b}\right)S = T_s^{mod} - \left[\alpha T_e \ln\left(\frac{pCO_2}{pCO_2^{mod}}\right) + f(pCO_2)\right]. \quad (7)$$

If $f(pCO_2) \propto \ln(pCO_2)$, which is the case for $pCO_2 \leq 0.1$ bar[43,44], then $S \propto -\ln(pCO_2)$. However, even if $f(pCO_2)$ deviates from log-linearity with p$CO_2$, $S$ will become increasingly log-linear with p$CO_2$ as $\alpha T_e$ increases. In Eq. (7), increasing $\alpha T_e$ will cause the $\ln(pCO_2)$ term to dominate the relationship between $S$ and p$CO_2$. Intuitively, increasing $\alpha T_e$ decreases the temperature dependence of continental weathering relative to its p$CO_2$ dependence. Note that bigger $T_e$ reduces the temperature

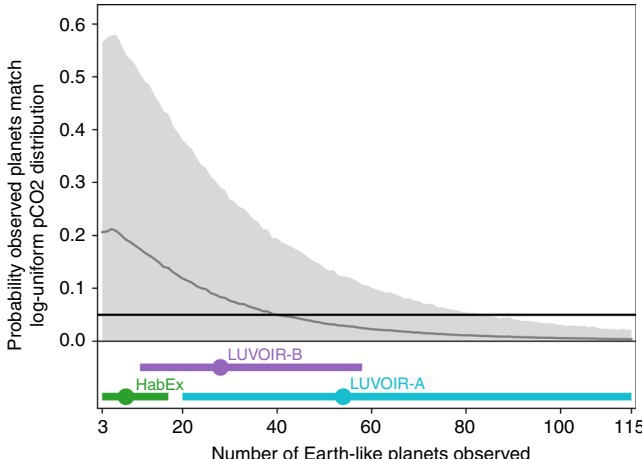

**Fig. 3 The probability observed exoplanets will accidentally match a log-uniform distribution for pCO$_2$ in the HZ if the true pCO$_2$ distribution is regulated by the carbonate–silicate weathering cycle, as shown in Fig. 1.** This probability is shown on the vertical axis. The horizontal axis shows the number of observed Earth-like exoplanets. The solid gray curve and corresponding shaded gray region show the mean probability and 2$\sigma$ uncertainty, respectively, that the observed planets, sampled from the planets shown in Fig. 1, match a log-uniform pCO$_2$ distribution in the HZ. This curve is calculated from 10,000 two-dimensional Kolmogorov–Smirnov tests (see Results, subsection "Observational uncertainty for pCO$_2$ in the HZ"). The solid, horizontal black line highlights the 5% probability line. At the bottom of the figure, the labeled points and error bars show the number of Earth-like exoplanets the next generation of proposed space telescopes are expected to observe (telescope data in Table 2). The vertical scaling of the telescope points is arbitrary, only the horizontal position and extent of the 1$\sigma$ error bars is significant. To rule out a log-uniform pCO$_2$ distribution with 95% confidence, future telescopes would need to observe at least 83 Earth-like planets.

dependence of continental weathering while bigger $\alpha$ increases the pCO$_2$ dependence of continental weathering (Eq. (4)).

In addition to predicting a linear relationship between $\log(\text{pCO}_2)$ and $S$, the carbonate–silicate cycle implies that moving an Earth-like planet outward in the HZ will cause $T_s(S, \text{pCO}_2)$ to decrease. For increasing orbital distance, pCO$_2$ must increase for $T_s(S, \text{pCO}_2)$ to increase in the HZ. From Eq. (5), pCO$_2$ will be greater than pCO$_2^{\text{mod}}$ in such cases so $T_s(S, \text{pCO}_2)$ must be less than $T_s^{\text{mod}}$. This decrease in $T_s(S, \text{pCO}_2)$ as $S$ decreases is shown in Fig. 2. Physically, the power law dependence of weathering on pCO$_2$ can balance volcanic outgassing at lower surface temperatures in the outer HZ.

Figure 2 shows the approximately log-linear relationship between pCO$_2$ and $S$ for the modern Earth moved outward in the HZ. The gray lines and colored circles in Fig. 2 show the expected pCO$_2$ value for the given incident flux $S$, calculated from Eq. (6). For each $S$ value in Fig. 2, Eq. (6) was solved for pCO$_2$ by using Eq. (8), the polynomial fit for surface temperature based on a 1D climate model (described in the Methods, subsection "Habitable zone 1D climate model"), assuming values of $\alpha T_e$.

The value of $\alpha T_e$ affects the slope of the relationship between $S$ and pCO$_2$ due to the carbonate–silicate weathering cycle, shown in Fig. 2. From above, the ranges for $\alpha$ and $T_e$ are $0.1 \leq \alpha \leq 0.5$ and $10 \leq T_e \leq 40$ (ref. [21]), so $1 \leq \alpha T_e \leq 20$. If we consider uniform distributions of $\alpha$ and $T_e$, then roughly 95% of $\alpha T_e$ values will be greater than 2.3. If $\alpha = 0.23$ and $T_e = 10$ K then $\alpha T_e = 2.3$, which is used for the Strong T-dep. curve in Fig. 2. The mean of each parameter, $\alpha = 0.3$ and $T_e = 25$ K gives $\alpha T_e = 7.5$, which corresponds to the Moderate T-dep. curve in Fig. 2. For $\alpha T_e \leq$

2.3 the colored points and gray curves become increasingly similar to the constant 289 K surface temperature contour in Fig. 2. However, for uniform distributions of $\alpha$ and $T_e$, ~95% of $\alpha T_e$ values are greater than 2.3, so an approximately log-linear relationship between $S$ and $\log(\text{pCO}_2)$ is the default expectation for Earth-like planets in the HZ.

**Observational uncertainty for pCO$_2$ in the HZ.** In the log-linear fit shown as the solid red line in Fig. 1, which is the expected relationship between pCO$_2$ and $S$ that we have derived above, the $r^2$-value is 0.49. Thus, about half the variance in $\log(\text{pCO}_2)$ is described by changes in incident flux. The slope is $3.92 \pm 0.24$ (95%) with units $-\log_{10}(\text{pCO}_2 \text{ [bar]})/(S/S_\oplus)$, so our model predicts a trend of increasing atmospheric CO$_2$ with orbital distance, which future missions might detect[32–34]. However, there is sufficient spread in our simulated planets that confirming the HZ hypothesis from such a trend may be difficult.

This difficulty is readily seen if we consider a log-uniform distribution for pCO$_2$ on Earth-like planets in the HZ. If we randomly generate 1050 such planets, where $10^{-4} \leq \text{pCO}_2 \leq 10$ bar is sampled log-uniformly, $0.35 \leq S \leq 1.05$ is sampled uniformly, and impose the same constraints on surface temperature for habitability as in Fig. 1, then the log-linear line of best fit through the uniform planet data has a slope of $3.76 \pm 0.465$ (95%) with units $-\log_{10}(\text{pCO}_2 \text{ [bar]})/(S/S_\oplus)$. Thus, measuring just the log-linear trend between pCO$_2$ and $S$ in the HZ is unlikely to test the HZ hypothesis as this measurement cannot confidently detect the presence of the carbonate–silicate weathering cycle—it is indistinguishable from that of randomly distributed pCO$_2$ between the surface temperature limits for habitability.

The inability to differentiate between the log-linear trends for weathering-mediated and random pCO$_2$ vs $S$ in the HZ is due to the assumed surface temperature constraints we impose in our model (between 248 and 355 K for planets in the HZ, see Methods, subsection "Numerical carbonate–silicate cycle modeling"). Such temperature constraints are necessary as the carbonate–silicate weathering cycle can only operate when water, as liquid, is present at the planetary surface. Even without the carbonate–silicate weathering cycle, a minimum surface temperature for habitable planets, which must exist, will result in increasing pCO$_2$ with orbital distance, as shown by the constant temperature contours in Fig. 1.

To test the HZ hypothesis, we propose searching for the two-dimensional (2D) distribution of planets in the $S$-pCO$_2$ phase space that arises from the carbonate–silicate weathering cycle. This $S$-$\log(\text{pCO}_2)$ relationship is shown by the point density in Fig. 1, where the distribution of habitable, stable planets is not log-uniformly distributed over pCO$_2$. Rather, around the best-fit line, there is an abundance of planets in the outer HZ at high pCO$_2$, a dearth of low pCO$_2$ planets between ~0.9 and ~0.7 $S/S_\oplus$, and few high-pCO$_2$ planets throughout the HZ compared to the log-uniform pCO$_2$ case. These differences are expected features of the carbonate–silicate weathering cycle due to the temperature- and pCO$_2$-dependent nature of the weathering feedback. Recall from section "Habitable zone climate theory revisited" that, as $S$ decreases, the lowered temperature will reduce weathering causing pCO$_2$ to increase. This results in the lack of low-pCO$_2$ planets in the middle of the HZ and the high abundance of habitable planets in the outer HZ (purple shaded region in Fig. 1). Similarly, for large pCO$_2$, the temperature is warmer and pCO$_2$ higher than that of modern Earth so the carbonate–silicate weathering cycle acts to lower pCO$_2$, which reduces the number of high-pCO$_2$ planets throughout the HZ relative to the outer HZ.

To detect the prevalence of the carbonate–silicate weathering cycle and test the validity of the HZ concept, future observations

**Table 2 The number of expected Earth-like exoplanets observed by each platform from the HabEx[30] and LUVOIR[31] final reports.**

| Telescope | Diameter (m) | Expected yield (1$\sigma$) |
|-----------|--------------|----------------------------|
| HabEx | 4 | $8^{+9}_{-5}$ |
| LUVOIR-B | 8 | $28^{+30}_{-17}$ |
| LUVOIR-A | 15 | $54^{+61}_{-34}$ |

should measure the 2D $S$-$pCO_2$ distribution of habitable, Earth-like exoplanets. This measured distribution can be compared to the distribution of habitable planets we predict in Fig. 1 to determine if Earth-like planets in the HZ are consistent with the $S$-$pCO_2$ predictions of the carbonate–silicate weathering cycle.

A test of the 2D phase space of $S$ and $pCO_2$ in the HZ is shown in Fig. 3, which was produced using a 2D Kolmogorov–Smirnov (KS) test. The 2D KS test compares the statistical similarity of a sample distribution to a reference distribution[47–49]. For Fig. 3, the reference distribution was comprised of 500 randomly generated planets from the log-uniform distribution for $pCO_2$ described above ($10^{-4} \leq pCO_2 \leq 10$ bar, $0.35 \leq S \leq 1.05 S/S_\oplus$, and surface temperature between 248 and 355 K). The sample distribution was generated by randomly selecting a number of planets from Fig. 1 equal to the number of observed exoplanets. For a given number of observed exoplanets in Fig. 3, the horizontal axis, we ran the KS test 10,000 times then calculated the mean and standard deviation from those runs, shown by the gray contour and shaded region. This resampling is necessary as the 2D KS test is a nonparametric approximation that two data sets come from the same underlying population[48]. We note that below ~20 observed planets and for probabilities above ~0.1, the 2D KS test used here can be unreliable[49]. These limitations do not invalidate the analysis shown in Fig. 3, as we want to know, with 95% confidence, that a log-uniform $pCO_2$ distribution can be ruled out if real exoplanets follow the distribution shown in Fig. 1, which corresponds to the gray line and shaded contour dipping below the 0.05 probability value, shown by the horizontal black line, at 83 observations in Fig. 3.

Thus, confidently detecting the carbonate–silicate weathering cycle will require many exoplanet observations, as shown in Fig. 3. Proposed NASA telescopes, HabEx and LUVOIR, are expected to observe between 3 and 115 Earth-like exoplanets[30,31] (see Table 2). The ranges for each mission concept are shown by the colored circles with error bars in Fig. 3. Only the nominal capability of LUVOIR-A, the variant of the proposed LUVOIR space telescope with a primary mirror diameter of 15 m, would provide sufficient Earth-like exoplanet detections to confidently discriminate between a log-uniform $pCO_2$ distribution in the HZ and a $pCO_2$ distribution regulated by the carbonate–silicate weathering cycle. A caveat is that this calculation does not consider the instrument uncertainty in derived $pCO_2$ measurements for each telescope or that other processes not considered in our model may alter $pCO_2$ in the HZ, as discussed below.

## Discussion

Our model assumes that the full variation and uncertainty in Earth's carbon cycle parameters through time (Table 1) are representative of habitable Earth-like exoplanets generally. This assumption is a reasonable first-order approximation as the bulk composition and geochemistry of rocky exoplanets appear similar to Earth's[50]. However, the validity of this assumption likely depends on the parameter in question. For example, it is probably reasonable to expect habitable exoplanets to have a wide range of land fractions and outgassing fluxes, but it is unclear whether

there is as much natural variability in the temperature dependence of silicate weathering. An improved mechanistic understanding of weathering on Earth[51,52] might reduce these uncertainties.

Other weathering feedbacks have been proposed to operate on the Earth through time, such as reverse weathering[53]. In reverse weathering, cations and dissolved silica released from silicate weathering are sequestered into clay minerals rather than carbonates so that $CO_2$ remains in the atmosphere, warming the climate and reducing ocean pH. Reverse weathering is thought to be strongly pH dependent and as ocean pH decreases, reverse weathering turns off, acting as a climate stabilization mechanism similar to the carbonate–silicate cycle. The importance of reverse weathering is so poorly constrained through Earth's history[45] that it does not make sense to consider it in our model. However, with future constraints from geology and lab measurements, reverse weathering might alter the stable $CO_2$ abundances of our modeled atmospheres shown in Fig. 1.

At both the inner and outer edges of the HZ, our model assumes that abundant liquid water exists at the planetary surface because, without a liquid surface ocean, the carbonate–silicate weathering cycle ceases and $CO_2$ cannot be sequestered after outgassing. Beyond these temperature bounds, other processes must regulate $pCO_2$. This is a caveat to consider in future observations. As we see from Fig. 1, Mars has low atmospheric $CO_2$ and low incident flux. Frozen exoplanets similar to Mars, populating the white area under the 248 K contour in Fig. 1, could exist in exoplanet surveys. Similarly, planets devoid of surface water, such as Venus, might exist at high $pCO_2$ within the HZ. If future observations detect such planets without confirming the existence of a liquid surface or surface temperature, it could introduce additional uncertainties in any relation between orbital distance and atmospheric $CO_2$. Detecting a surface ocean, one of the most important surface features to confirm when searching for biosignatures and habitability[54–56], is also important to interpret trends of $CO_2$ in the HZ.

Because we only consider variations on an Earth-like planet, our model predictions may underestimate the inherent variability in habitable exoplanetary conditions. Planets very different from the modern Earth, such as waterworlds without a carbonate–silicate weathering cycle[57] or $CH_4$-rich worlds[58,59], could introduce additional uncertainty in an observed relationship between $S$ and $pCO_2$ in the HZ. Despite such uncertainties, future missions should measure the relationship between $S$ and $pCO_2$ in the HZ, or possibly a sharp transition in $pCO_2$ at the inner edge of the HZ due to loss of surface water and subsequent shutoff of surface weathering[38,60]. A more complex model than presented here is necessary to predict such a jump in $pCO_2$ at the inner edge of the HZ. However, if the carbonate–silicate weathering cycle is indeed ubiquitous, as is typically assumed in HZ calculations, then the relationship between incident flux and $pCO_2$ may follow the $S$-$pCO_2$ relationship predicted in Fig. 1. If no such relationship is observed, then the carbonate–silicate weathering cycle may have limited influence on planetary habitability and the limits of the conventional HZ could need revision. Alternatively, the HZ hypothesis could be incorrect and the long-term climate of HZ planets could be set by phenomena beyond those considered here.

A previous version of this work was published as part of a Ph. D. thesis[61].

## Methods

**Habitable zone 1D climate model.** We use the Virtual Planetary Laboratory (VPL) 1D radiative-convective climate model[5,62] to generate surface temperatures for an Earth-like planet at various $pCO_2$ and incident fluxes. We consider incident fluxes between $1.05 S_\oplus$ and $0.35 S_\oplus$, the HZ limits for a Sun-like star[6], and

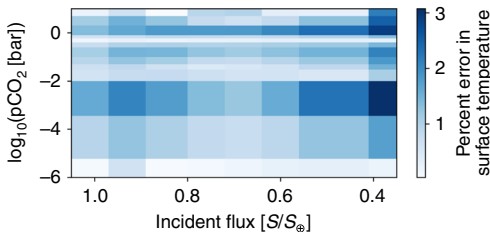

**Fig. 4 The relative error between our fourth-order polynomial fit and the full 1D radiative-convective climate model.** Our polynomial fit is valid between $1.05 S_\oplus$ and $0.35 S_\oplus$, where $S_\oplus$ is the solar constant. The polynomial fit is valid for atmospheric $CO_2$ abundances between $10^{-6}$ and 10 bar. The surface temperatures predicted by the polynomial fit reproduce the results of the 1D climate model. The maximum error in predicted surface temperature between the polynomial fit and the 1D climate model is ~3%.

atmospheric $CO_2$ partial pressures between $10^{-6}$ and 10 bar. We assume the atmosphere is comprised of $CO_2$ and $H_2O$. If the $CO_2$ partial pressure is below 1 bar, we set the initial atmospheric pressure to 1 bar and add $N_2$ to the atmosphere such that the total surface pressure is 1 bar. We fix the stratospheric water vapor concentration to the modern Earth value and follow the Manabe–Wetherald relative humidity distribution in the troposphere with empirical constraints based on the modern Earth[63].

We fit the surface temperature output, $T_s$ in K, from the climate model with a fourth-order polynomial in $\ln(pCO_2)$ and normalized stellar flux, as follows:

$$
\begin{aligned}
T_s(S, pCO_2) = {} & 4.809 - 222.0X - 68.44X^2 - 6.737X^3 - 0.206X^4 \\
& + 1414XY + 446.4X^2Y + 44.41X^3Y + 1.364X^4Y \\
& - 2964XY^2 - 978.4X^2Y^2 - 98.86X^3Y^2 - 3.059X^4Y^2 \\
& + 2655XY^3 + 907.5X^2Y^3 + 92.87X^3Y^3 + 2.892X^4Y^3 \\
& - 868.4XY^4 - 304.6X^2Y^4 - 31.48X^3Y^4 - 0.985X^4Y^4 \\
& + 1045Y - 1496Y^2 + 1064Y^3 - 281.1Y^4.
\end{aligned}
\tag{8}
$$

Here, $CO_2$ partial pressure $pCO_2$ is in bar, $X = \ln(pCO_2)$, and $Y = S/S_\oplus$ is the incident flux, $S$, normalized to the solar constant, $S_\oplus$. Figure 4 shows the agreement between the 1D climate model and the polynomial fit used in this work.

**Numerical carbonate–silicate cycle modeling.** To calculate the steady-state $pCO_2$ in the atmospheres of Earth-like planets in the HZ, we use a weathering model that describes $pCO_2$ on the Earth through time[21,26]. We summarize the model below and highlight how the model in this work differs from previous implementations[21,26]. These previous implementations provide a comprehensive explanation and justification of the model parameterizations, and empirical and theoretical basis. The model, as a Python script, is available in the Supplementary Data and contains a complete description of the model equations and parameters (see the file weathering_model.py).

The weathering model balances the flux of outgassed $CO_2$ against the loss of carbon due to continental and seafloor weathering, which result in precipitation of carbonates in the ocean and seafloor pore space. Quantitatively, for time $t$, this is described by time-dependent equations where we normalize to the mass of the ocean, $M_o$ (nominally, an Earth ocean, $1.35 \times 10^{21}$ kg):

$$
\begin{aligned}
\frac{dC}{dt} &= \frac{F_{out} + F_{carb} - P_o - P_p}{M_o} \\
\frac{dA}{dt} &= 2 \times \frac{F_{sil} + F_{carb} + F_{diss} - P_o - P_p}{M_o}.
\end{aligned}
\tag{9}
$$

Here, $C$ is the non-organic carbon content of the atmosphere–ocean system in mol C kg$^{-1}$, and $A$ is the carbonate alkalinity in mol equivalents (mol eq). Carbonate alkalinity (henceforth alkalinity) is the charge-weighted sum of the mol liter$^{-1}$ concentration of bicarbonate and carbonate anions, $[HCO_3^-] + 2[CO_3^{2-}]$. $F_{out}$ is the global $CO_2$ outgassing flux, $F_{carb}$ and $F_{sil}$ are the continental carbonate and silicate weathering fluxes, $F_{diss}$ is the rate of seafloor basalt dissolution, and $P_p$ and $P_o$ are the pore and ocean precipitation fluxes. The fluxes on the right-hand side of Eq. (9) ($F_{out}$, $F_{carb}$, $P_o$, $P_p$, $F_{sil}$, $F_{carb}$, and $F_{diss}$) are given in mol C yr$^{-1}$ for $dC/dt$ and in mol eq yr$^{-1}$ for $dA/dt$.

The alkalinity that enters the ocean from weathering will balance a $+2$ charge cation (e.g., $Ca^{++}$), which is why a factor of 2 enters in the definition of $dA/dt$ in Eq. (9). Hence, geochemists often think of alkalinity in terms of the balance of cations produced in weathering, principally $Ca^{++}$. This reasoning arises because the weighted sum of carbonate and bicarbonate concentrations must balance the charge of conservative cations minus conservative anions (i.e., $2[Ca^{++}] + 2[Mg^{++}] + Na^+ + \ldots - [Cl^-] - \ldots$), ignoring minor contributions from weak acid anions and water dissociation. Weathering releases

cations and carbon speciation adjusts to ensure charge balance, so that the cation release is effectively equivalent to carbonate alkalinity.

To improve the rate of model convergence and range of model inputs over which Eq. (9) converges, we do not consider the seafloor pore space and atmosphere–ocean as separate systems. This differs from previous versions of the model[21], which considered the atmosphere–ocean and pore space independently. Rather, we approximate the atmosphere–ocean and pore space as a single entity in Eq. (9). This simplification does not appreciably change the model output for atmospheric $CO_2$ because we run the model to steady state in all cases, where the atmosphere–ocean and pore space reach approximate equilibrium. In the next section, we present additional details on our model implementation and discuss the agreement between our no-pore model and the original, two-box model[21].

A second modification is the range of incident stellar fluxes over which the model can be run. Previously, the model described here was used to study the Earth through time[21] and thus only considered solar fluxes between $S_\oplus$ (the modern solar constant) and early Earth's $0.7 S_\oplus$ ($S_\oplus = 1360$ W m$^{-2}$). We extend to include the entire conservative HZ of a Sun-like star, roughly $1.05 S_\oplus$ to $0.35 S_\oplus$[6]. We use Eq. (8), the fourth-order polynomial fit to a 1D climate model, to calculate surface temperatures throughout the HZ. The Bond albedo of the planet is calculated dynamically by the climate model and thus included implicitly in our polynomial fit.

With the coupled climate and weathering model, we generate steady-state, Earth-like planets by randomly sampling plausible initial model inputs. The ranges for each parameter we consider are representative of the Earth through time[21] and shown in Table 1. These ranges represent very broad uncertainties of the carbonate–silicate cycle on the Earth through time and so are appropriate for Earth-like planets. We conservatively assume a uniform distribution for each parameter range shown in Table 1.

We parameterize the internal heat of an Earth-like planet conservatively using the planet's age, ranging 0–10 Gyr, which is the approximate habitable lifetime of an Earth-like planet around a Sun-like star[64]. The equation for planetary heat relative to the modern Earth, $Q$, is given by

$$
Q = \left(1 - \frac{4.5 - \tau}{4.5}\right)^{-n_{out}},
\tag{10}
$$

where $\tau$ is the age of the planet in Gyr, and $n_{out}$ is the scaling exponent for internal heat, with a range given in Table 1.

The parameter ranges shown in Table 1 represent the uncertainty of the carbonate–silicate weathering cycle on the Earth through time[21]. Implicit in our assumed parameter ranges is that continental land fraction, $f_{land}$, and biological weathering fraction, $f_{bio}$, have increased from 0 when the Earth formed to 1 on the modern Earth. Similarly, the relative internal heat, $Q$, is assumed to be large when the Earth is young and unity for the modern Earth. Therefore, on the modern Earth, where $f_{land} = 1$, $f_{bio} = 1$, and $Q = 1$, the weathering rate is maximized and outgassing rate is relatively small (see Methods, subsection "Validity of carbon cycle parameterizations to exoplanets" for a discussion on the importance of these three parameters in our model). This is seen in Fig. 1, where the modern Earth appears near the lower bound for predicted $pCO_2$ in the HZ. If the continents on an exoplanet were more easily weathered or outgassing much lower than on the modern Earth, such exoplanets could have $pCO_2$ values well below the modern Earth value shown in Fig. 1. We do not consider such exoplanets in this model, so the results presented here are only applicable to planets similar to the Earth through time.

Our model assumes that each simulated planet is habitable, i.e., it has a stable, liquid surface ocean, a necessity for the carbonate–silicate cycle to operate. For a mean surface temperature below 248 K, Earth-like planets are likely completely frozen[36], which we use as a lower temperature bound in the model. While 248 K is below the freezing point of water, it is a global mean surface temperature and 3D models show that the range 248–273 K for this parameter does not preclude the existence of a liquid ocean belt near the equator. At the other temperature extreme, a hot, Earth-like planet can rapidly lose its surface oceans due to high atmospheric water vapor concentrations that are photolyzed and subsequently lost to space. This upper temperature bound on habitability occurs at ~355 K[37]. Above 355 K, Earth-like planets are unlikely to remain habitable for more than ~1 Gyr[37] and cannot operate a carbonate–silicate cycle over geologic timescales. We use these two temperature bounds, 248 K and 355 K, as the limits for habitability in our model. Any modeled planet with a final surface temperature outside these limits is uninhabitable and removed from our results.

We limit HZ planets to those with $pCO_2$ below 10 bar. For most Earth-like planets in the HZ, 10 bar of $CO_2$ results in planets with surface temperatures well above 355 K, which are not habitable on long time scales. If we impose a fixed stratospheric water vapor concentration in the 1D climate model and modify the tropospheric water vapor concentration based on empirical data from the modern Earth, we enable the 1D climate model to accurately model habitable, Earth-like planets through much of the HZ. But in the outer HZ, with more than ~10 bar of $CO_2$, this assumption overestimates atmospheric water vapor concentrations and leads to artificially warm planets, so we reject such cases. Above ~10 bar of $CO_2$ in the outer HZ, assuming a saturated troposphere for water vapor, increasing atmospheric $CO_2$ may not lead to additional warming[6]. Rather, the surface cools in such scenarios because additional $CO_2$ leads to increased Rayleigh scattering and no additional warming. Because Earth-like planets in the outer HZ would be frozen

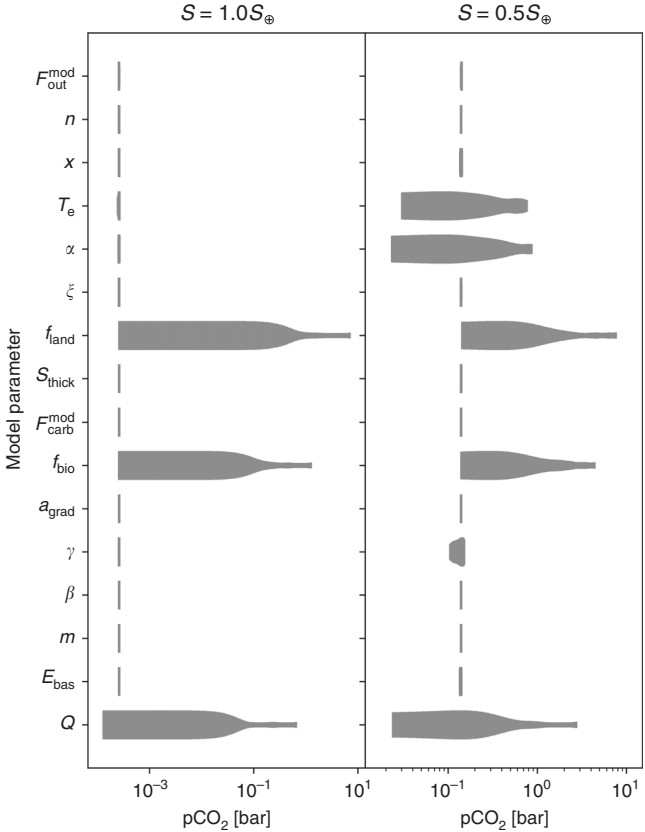

**Fig. 5 The spread in steady-state pCO$_2$ from varying a single model parameter.** Each parameter in Table 1 is shown on the vertical axis (note that $n_{out}$ and $\tau$ are incorporated into Q, see Eq. (10)). The left panel shows an incident flux of $S = 1.0S_\oplus$. The right panel shows an incident flux of $S = 0.5S_\oplus$. For each parameter, we held all other parameters constant at the modern Earth value (see text) and randomly sampled 100 values for the parameter in question from uniform distributions of the ranges given in Table 1. The horizontal extent of the gray shaded region shows the range of possible pCO$_2$ values when all other parameters are fixed. The thickness of each gray shaded region shows the relative abundance of steady-state planets at that pCO$_2$. The thickest regions show maximum relative abundance, the thinnest regions show the minimum relative abundance. No surface temperature limits on habitability were imposed for the simulated planets. At low pCO$_2$, three parameters ($f_{land}$, $f_{bio}$, and Q) dominate the spread in pCO$_2$. At higher pCO$_2$, the temperature- and pCO$_2$-dependence of continental silicate weathering ($T_e$ and $\alpha$) are also important.

and uninhabitable even with CO$_2$ partial pressures above ~10 bar, we impose a 10-bar limit for CO$_2$ in the outer HZ. This limit agrees with previous CO$_2$ limitations in coupled climate and weathering models[29].

**Combined ocean and pore space model justification.** The carbon cycle model used in this work was previously derived as a two-box model[21], where the atmosphere–ocean and the seafloor pore space were separated. In this work, we combine the ocean–atmosphere and the pore space into a single unit. This modification can be implemented in the original model[21] by assuming that the pH of the pore space is the same as the pH of the ocean, and assuming that the alkalinity and carbon content of the ocean and pore space are the same. The dissolution and precipitation fluxes can then be calculated without treating the ocean–atmosphere and the pore space as different systems. This modification allows the model to converge quicker over a wider range of parameter combinations.

To validate our combined model, we ran the modern Earth through both the original, two-box model[21] and our modified model at ten different incident fluxes between $S_\oplus$ and $0.7S_\oplus$. The average error in predicted CO$_2$ values between our model and the two-box model was 2.8%, with a minimum error of 2.3%, and a maximum error of 3.6%. Given the large uncertainties in model inputs (Table 1), the few percent error introduced by our simplified model is unimportant.

For each parameter combination in our simplified model, we start with the modern Earth then impose a step change for each model parameter. We then run

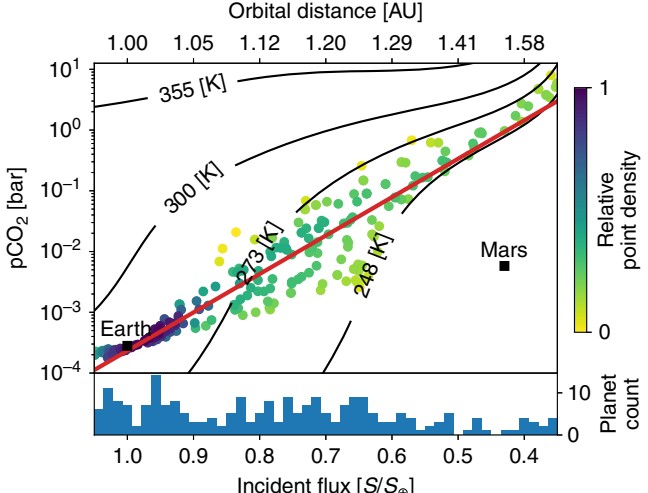

**Fig. 6 The expected distribution of stable, steady-state pCO$_2$ on Earth-like planets if Q, $f_{land}$, and $f_{bio}$ are fixed to 1, i.e., modern Earth values.** Except for fixing $Q = f_{land} = f_{bio} = 1$, this figure is generated identically to Fig. 1. The spread in pCO$_2$ in the outer HZ is due to the temperature- and pCO$_2$-dependence of continental weathering ($T_e$ and $\alpha$). This is expected from Eq. (4), which shows that $T_e$ and $\alpha$ will be increasingly influential as pCO$_2$ and surface temperature deviate from the modern Earth values, as discussed in the Results, subsection "Habitable zone climate theory revisited". Without changes in Q, $f_{land}$, and $f_{bio}$, there is little spread in pCO$_2$ in the inner HZ.

the simulation for 10 Gyr or until the system reaches steady state. We consider the model to have reached steady state when extrapolation of the rate of change of pCO$_2$ for 1 Gyr changes pCO$_2$ by <1%. Typically, the model converges within a few Myr to a few tens of Myr. Rarely (2 of the 1200 planets simulated in this work), parameter combinations will not reach steady state after 10 Gyr. Simulations with combinations of exceptionally high outgassing rates and low CO$_2$ weathering rates can enter a regime were atmospheric CO$_2$ builds without bound, never converging. Such model results are beyond the range of validity of our model.

**Validity of carbon cycle parameterizations to exoplanets.** The parameterization of weathering in our model has been empirically validated for the modern Earth[16,21,26]. The exponential temperature dependence of continental weathering is a reasonable approximation that agrees with field and lab measurements[16] and can reproduce the climate results of more complex models[21,26]. Similarly, the power-law parameterization for the pCO$_2$ dependence of continental weathering agrees with data from the modern Earth[26] and can even be approximately derived from equilibrium chemistry arguments for an Earth-like exoplanet[38]. The bulk geochemistry of rocky exoplanets may be similar to Earth's[50], so we expect our weathering parameterization to reasonably approximate Earth-like planets in the HZ. However, uncertainties in how the carbonate–silicate weathering cycle regulates climate on Earth persist[21], so the predicted variations in pCO$_2$ in our model may not capture the true variability of pCO$_2$ in the HZ. Below, we show that our broad parameterization of the carbonate–silicate weathering cycle may encompass the plausible range of pCO$_2$ for the Earth through time, but improved understanding the carbonate–silicate weathering cycle may be necessary to know if such variations are indeed representative of the Earth through time and applicable to Earth-like planets generally.

The rate of weathering depends strongly on the intrinsic features of a planet, such as the CO$_2$ outgassing rate and the properties of its continents. Changes in continental uplift rate, lithology, and configuration are parameterized in our model through the $f_{land}$ and $f_{bio}$ terms. The parameters $f_{land}$ and $f_{bio}$ linearly scale the weathering flux and could analogously be considered a continental weatherability scaling factor. For the ranges of $f_{land}$ and $f_{bio}$ considered in our model (see Table 1), changes in the continental weatherability alone can generate pCO$_2$ values spanning ~4 orders of magnitude. This broad parameterization likely encompasses pCO$_2$ perturbations due to continental weatherability changes caused by large volcanic eruptions or changes in continental configuration. Indeed, the largest, constrained change in pCO$_2$ due to such events on Earth may be closer to ~1 order of magnitude, coeval with the eruption of the Siberian Traps[65].

The importance of continental weatherability ($f_{land}$ and $f_{bio}$) on pCO$_2$, relative to other parameters, is shown in Fig. 5. Figure 5 was generated by sampling uniform distributions for each model parameter shown in Table 1 across its listed range. When one parameter was varied, all other parameters were held constant at their modern Earth value, which we define as: $F_{mod}^{out} = 6$ Tmol C yr$^{-1}$, $n = 1.75$, $x = 1$, $T_e = 25$ K, $\alpha = 0.3$, $\xi = 0.3$, $f_{land} = 1$, $S_{thick} = 1$, $F_{carb}^{mod} = 10$ Tmol C yr$^{-1}$,

$f_{bio} = 1$, $a_{grad} = 1.075$, $\gamma = 0.2$, $\beta = 0.1$, $m = 1.5$, $E_{bas} = 90$ kJ mol$^{-1}$, and $Q = 1$. Note that we incorporate $n_{out}$ and $\tau$ from Table 1 into $Q$, the internal heat (see Eq. (10)), which is the parameter of interest. We show two different values for $S$ in Fig. 5, $S = 1.0S_{\oplus}$ in the left panel and $S = 0.5S_{\oplus}$ in the right panel. For both values of $S$, Fig. 5 shows that variations in $f_{land}$ and $f_{bio}$ alone can alter pCO$_2$ by orders of magnitude.

The internal heat of the planet, $Q$, plays a similarly important role in setting pCO$_2$. The rate of CO$_2$ outgassing is determined by $Q$ and our broad parameterization of $Q$ allows pCO$_2$ to vary by orders of magnitude throughout the HZ, as shown in Fig. 5.

The rate of CO$_2$ outgassing and continental weatherability drive the majority of the spread in pCO$_2$ shown in Fig. 1. This is readily seen in Fig. 6, which shows the results of 300 random parameter combinations from uniform distributions of the parameters in Table 1 except for $Q$, $f_{land}$, and $f_{bio}$, which were all fixed to 1. Of the 300 parameter combinations, 235 remained above 248 K and are shown in Fig. 6. Comparing Fig. 6 to Fig. 1, it is readily apparent that the broad uncertainty in pCO$_2$ from our results is due to variations in intrinsic planetary properties ($Q$, $f_{land}$, and $f_{bio}$) rather than uncertainties in the tuning parameters of our carbon cycle parameterization.

The outgassing rate and continental properties of habitable exoplanets remain unknown. Thus, our broad parameterization of those terms, which align with possible conditions on Earth throughout its history, are a reasonable approximation. If an Earth-like, carbonate–silicate weathering cycle is common on habitable planets, then these parameters may largely determine pCO$_2$ on such planets and generate a range for pCO$_2$ at a given orbital distance similar to that shown in Fig. 1.

## Data availability
The data used in this work are available in the Supplementary Data. Our model code depends on the location of the data directory, so the data and model are provided together in a single, zipped file.

## Code availability
The code used to generate the data and figures for this work is available in the Supplementary Data.

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

## Acknowledgements

We would like to thank Nicholas Wogan for his constructive suggestions on our initial manuscript. We also thank NASA's Virtual Planetary Laboratory (grant 80NSSC18K0829) at the University of Washington and the NASA Pathways Program for funding this work. J.K.T. was supported by NASA through the NASA Hubble Fellowship grant HF2-51437 awarded by the Space Telescope Science Institute, which is operated by the Association of Universities for Research in Astronomy, Inc., for NASA, under contract NAS5-26555.

## Author contributions

O.R.L., D.C.C., and J.K.T. all contributed to the theoretical and conceptual aspects of this work and the drafting of the manuscript. O.R.L. implemented the numerical model and generated model data.

## Competing interests

The authors declare no competing interests.
