## [Peer Review File · Nature Communications]

Editorial Note: This manuscript has been previously reviewed at another journal that is not operating a transparent peer review scheme. This document only contains reviewer comments and rebuttal letters for versions considered at *Nature Communications*. Mentions of the other journal have been redacted.

REVIEWER COMMENTS

Reviewer #1 (Remarks to the Author):

Review of “Carbonate-Silicate Cycle Predictions of Earth-like Planetary Climates and Testing the Habitable Zone Concept”

This is a revision of a manuscript previously submitted to [redacted]. The revision clarifies some issues from the original submission, and answers much of the comments from my earlier review. In particular, rearranging the results to show the full model results first (new figure 1), then going into the theory for how the S vs $\log(p\text{CO}_2)$ relationship arises makes things much cleaner. And I now get why the authors go through the exercise in section 2.2, as developing the functional form a trend of increasing $p\text{CO}_2$ with decreasing S would be expected to take is needed for testing whether the carbonate-silicate cycle does act as a stabilizing feedback for exoplanets. Overall this paper makes a nice contribution to our understanding of planetary habitability and how future observations can test models of climate regulation via the carbonate-silicate cycle. There are some important outstanding issues that need to be addressed with additional revision, however, but after this the paper will be suitable for publication in *Nature Communications*. In particular there is a significant bias in the proposed methodology for testing the carbonate-silicate cycle climate stabilization hypothesis that needs to be addressed. Furthermore, additional consideration of how well the models really cover the range of potential climate states we might expect to exist in nature, even when restricting the analysis to Earth-like planets, is needed. While both issues are important, especially the potential bias in the carbonate-silicate cycle hypothesis test, I think all my comments can be addressed by the authors without too much additional work.

1. The main finding of the paper is estimating the trend in atmospheric CO_2 versus stellar radiation one would expect if the carbonate-silicate cycle is active and stabilizing climate, and the scatter around this trend we'd expect based on uncertainties in how the carbonate-silicate cycle works. The paper lays out how many planets would need to be observed to confirm this trend of increasing CO_2 with orbital distance, and therefore test the hypothesis that the carbonate-silicate cycle stabilizes climate. However, the paper actually does not lay out this hypothesis and how it would be tested

particularly clearly, and based on my understanding, the methodology advocated introduces an important bias that could potentially invalidate any such hypothesis test.

The paper uses the results of their carbonate-silicate cycle model to establish an expected relationship between CO₂ and incident flux/orbital distance, if the carbonate-silicate cycle does act to stabilize climate. It is then implied (though not clearly stated) that observing this trend on real exoplanets would be a test of the carbonate-silicate cycle climate regulation hypothesis. However, the authors throw out all model results that produce “non-habitable” climates in determining this expected S-CO₂ trend. It is further implied (this is another area where the manuscript is quite unclear) that in trying to observe this trend, one would also throw out any observed planets with non-habitable climates (below 248 K, above 355 K). I raised this issue in my previous review about how this would influence the estimates for the number of planets that would need to be observed to detect the expected S-CO₂ trend. However, looking at the manuscript again I realized this is a much more important issue; following the methodology of only considering planets in the temperature range of 248-355 K, one could observe a trend of increasing CO₂ with decreasing S or increasing orbital distance, even if there is no climate regulation at all! This trend would arise just because a planet needs more CO₂ to have a temperate climate when the absorbed radiation is lower (and other greenhouse gases are not considered).

A potential null hypothesis to the idea that the carbonate-silicate cycle regulates atmospheric CO₂ levels is that atmospheric CO₂ on exoplanets will be totally random. In this case there would be points randomly scattered all over Figure 1. If we then throw out all the planets that are colder than 248 K and hotter than 355 K, we are left with a band of data points between these temperature contours, where CO₂ increases with orbital distance. One could fit a trend line to these remaining planets and find a similar CO₂-incident flux relationship to the one proposed in the paper, all without any actual climate regulation! Observing a trend of increasing CO₂ with decreasing incident flux among just planets with temperate climates is not a test of the carbonate-silicate cycle climate regulation hypothesis, as climate regulation is not required to produce this trend.

The paper needs to more clearly lay out their hypothesis for carbonate-silicate cycle climate regulation, and how it would be tested, in light of this issue. A better test might be the percentage of planets that end up with habitable versus non-habitable climates; if there is active climate regulation we'd expect a much higher proportion of planets to end up in the temperate climate range than if there is no climate regulation, and atmospheric CO₂ levels are totally random from planet to planet. The authors could then estimate how many planets would need to be observed to test these expectations. Another possibility is just considering all planets, even if they result in non-habitable climates, in both the models and observations. A trend of increasing CO₂ with decreasing incident flux among all planets observed in the habitable zone would serve as a test of whether climate regulation occurs. If we stick with just planets that have temperate climates, the CO₂-incident flux relationship for the random CO₂ case and a case with active climate regulation might have different functional forms or different slopes that could be potentially distinguished with observations. The

authors could then determine how many planets we'd need to observe to distinguish between these different CO₂-incident flux relationships.

2. Another key point is whether the models cover the full uncertainty range expected for exoplanets, or even just Earth-like exoplanets. The new discussion in the methods section and new figures 5 & 6 are really helpful here, in showing what factors drive uncertainty in the models. I appreciate that the authors are more careful now to state that they are assuming only Earth-like planets, and real variability could be greater. But I think the paper is still overconfident that they are even capturing the full variability for Earth-like planets. I don't think the authors need to modify their model; the approach taken is a reasonable first order estimate. But the text is far too dismissive of additional uncertainty on top of what they modeled, even for just "Earth-like" planets. Stating their conclusions in this way could then be potentially misleading to readers, especially those not as familiar with how uncertain our understanding of the carbonate-silicate cycle really is.

The main argument is that the model, primarily through variations in the outgassing rate, biological mediation of weathering, and land area, produces variations in CO₂ larger than observed during periods of major climate change in Earth's past. However it matters what factors actually lead to these climate swings Earth has experienced, as different climate swings have different proposed mechanisms: changes in "weatherability" due uplift and erosion (Kump & Arthur, 1997); changes in lithology, like exposing large areas of highly reactive mafic crust (McDonald et al 2019); changes in outgassing rate, and so on. Changes in lithology or erosion rate are not included in the modeling, but either factor can lead to significant changes in weathering rates on their own. The authors argue their land fraction and biological mediation factors basically cover the range of possible changes in "weatherability." However, lumping independent factors like erosion rate or lithology into one parameter means missing possible edge cases, where the independent factors all align at one extreme or the other, unless the catch-all parameter is properly calibrated to account for this. For example, if the authors included a weatherability factor, capturing variations in erosion rate due to uplift events, in addition to f_{land} and f_{bio} , there would be cases where all three of these parameters end up at their extrema, producing a greater spread in CO₂ than the current model predicts. Just because the model can produce large CO₂ variations doesn't mean it is capturing the full uncertainty range and possible edge cases that could arise. Given that we lack a complete mechanistic understanding of how the carbonate-silicate cycle operates (the governing equations of the model this paper uses could be missing significant factors, even beyond the ideas of "weatherability" I bring up here), it is really not justifiable to say that the model is capturing the full uncertainty range in climate states, even for only "Earth-like" planets.

3. Related to point 2, it is surprising that Earth sits at the lower bound of atmospheric CO₂, for the present solar flux, in Figure 1. How does sampling across all the different model parameters used not result in some cases with lower atmospheric CO₂ than Earth? The model predicts that the typical planet at Earth's incident flux would have ~100 times the atmospheric CO₂. Why is this?

4. At the end of section 4.3, the authors state that it can take ~ 1 Gyr for the models to reach a steady-state. This is surprising, as usually the weathering rate is expected to adjust and balance outgassing more rapidly, on the order of millions of years. So why do some models take so long to reach a steady-state? This is an important issue, because the model results in Figure 1 assume a steady-state in the carbonate-silicate cycle. Rates of volcanic outgassing change over hundreds of millions of years to Gyr timescales (e.g. Tajika & Matsui 1992, Foley & Smye 2018, Dorn et al 2018), so if similar timescales are needed for steady-state in the carbonate-silicate cycle to be established, then planets will be unlikely to actually be in such a steady-state. Atmospheric CO₂ levels on real planets could then be in a transient state, and be very different than the model predictions.

Minor comments:

Lines 73-77: This isn't stated quite right. It's not that any volcanic outgassing will lead to an increase in surface temperature and weathering rates. To cause an increase in surface temperature, and kick off the stabilizing feedbacks of the carbonate-silicate cycle as described, requires an increase in outgassing rate above the previously established steady-state between weathering and outgassing.

Equation 4: I don't understand how the exponent c here incorporates both silicate and carbonate weathering. Since carbonate weathering doesn't lead to net CO₂ drawdown, usually only silicate weathering is included in developing simple weathering models like the equations developed in section 2.2. Is equation 4, and subsequent equations, meant to incorporate both silicate and carbonate weathering? If so it would be more clear to write things as a sum of these two weathering equations, or at least spell this out more directly. Or if equation 4 is just the silicate weathering rate, then it would be more clear to just use α instead of defining the new parameter c .

Line 376: "discusses" should be "discuss"

Reviewer #2 (Remarks to the Author):

Revision of: Lehmer and co-workers "Carbonate-Silicate Cycle Predictions of Earth-like Planetary Climates and Testing the Habitable Zone Concept"

The authors have addressed the concerns and suggestions raised by the first round of review. The revised manuscript was a pleasure to read, and my comments below are aimed at clarifying the details of the model, which I think will increase the impact of this research.

Comments:

My reading of the literature is that the authors constrained their model by adopting the reasonable temperature range of 248-355 K. The obtained distribution of stable climate solutions presented in Figure 1 basically follows this, but there is also a void where no solution exists even at >248 K for an incident flux of $>\sim 0.65$. I am struggling with whether the authors have a mechanism whereby such solutions are precluded. My sense is no, and I am not sure why the model cannot find any solutions there. Is it simply because the parameter ranges in Table 1 do not explore such solutions? In other words, I'd like to understand what controls the lower boundary of the expected distribution of stable climates. It maybe that I am really missing the point here, but in this case this should be clarified. I think this would be an important thing to consider, because the authors' following arguments (e.g., Fig. 3) may be affected if the solutions exit there when the wider parameter ranges are explored.

I would recommend this manuscript for publication, but only after the authors discussed or dealt with above remaining concern.

Below are our responses to reviewer comments. The reviewer comments are in black and our responses are in this blue color. Changes in the manuscript are in this blue color as well.

Reviewer #1 (Remarks to the Author):

Review of “Carbonate-Silicate Cycle Predictions of Earth-like Planetary Climates and Testing the Habitable Zone Concept”

This is a revision of a manuscript previously submitted to [redacted]. The revision clarifies some issues from the original submission, and answers much of the comments from my earlier review. In particular, rearranging the results to show the full model results first (new figure 1), then going into the theory for how the S vs $\log(p\text{CO}_2)$ relationship arises makes things much cleaner. And I now get why the authors go through the exercise in section 2.2, as developing the functional form a trend of increasing $p\text{CO}_2$ with decreasing S would be expected to take is needed for testing whether the carbonate-silicate cycle does act as a stabilizing feedback for exoplanets. Overall this paper makes a nice contribution to our understanding of planetary habitability and how future observations can test models of climate regulation via the carbonate-silicate cycle. There are some important outstanding issues that need to be addressed with additional revision, however, but after this the paper will be suitable for publication in Nature Communications. In particular there is a significant bias in the proposed methodology for testing the carbonate-silicate cycle climate stabilization hypothesis that needs to be addressed. Furthermore, additional consideration of how well the models really cover the range of potential climate states we might expect to exist in nature, even when restricting the analysis to Earth-like planets, is needed. While both issues are important, especially the potential bias in the carbonate-silicate cycle hypothesis test, I think all my comments can be addressed by the authors without too much additional work.

1. The main finding of the paper is estimating the trend in atmospheric CO_2 versus stellar radiation one would expect if the carbonate-silicate cycle is active and stabilizing climate, and the scatter around this trend we'd expect based on uncertainties in how the carbonate-silicate cycle works. The paper lays out how many planets would need to be observed to confirm this trend of increasing CO_2 with orbital distance, and therefore test the hypothesis that the carbonate-silicate cycle stabilizes climate. However, the paper actually does not lay out this hypothesis and how it would be tested particularly clearly, and based on my understanding, the methodology advocated introduces an important bias that could potentially invalidate any such hypothesis test.

The paper uses the results of their carbonate-silicate cycle model to establish an expected relationship between CO_2 and incident flux/orbital distance, if the carbonate-silicate cycle does act to stabilize climate. It is then implied (though not clearly stated) that observing this trend on real exoplanets would be a test of the carbonate-silicate cycle climate regulation hypothesis. However, the authors throw out all model results that produce “non-habitable” climates in determining this expected S - CO_2 trend. It is further implied (this is another area where the manuscript is quite unclear) that in trying to observe this trend, one would also throw out any observed planets with non-habitable climates (below 248 K, above 355 K). I raised this issue in my previous review about how this would influence the estimates for the number of planets that would need to be observed to detect the expected S - CO_2 trend. However, looking at the manuscript again I realized this is a much more important issue; following the methodology of only considering planets in the temperature range of 248-355 K, one could observe a trend of increasing CO_2 with decreasing S or increasing orbital distance, even if there is no climate regulation at all! This trend would arise just because a planet needs more

CO₂ to have a temperate climate when the absorbed radiation is lower (and other greenhouse gases are not considered).

A potential null hypothesis to the idea that the carbonate-silicate cycle regulates atmospheric CO₂ levels is that atmospheric CO₂ on exoplanets will be totally random. In this case there would be points randomly scattered all over Figure 1. If we then throw out all the planets that are colder than 248 K and hotter than 355 K, we are left with a band of data points between these temperature contours, where CO₂ increases with orbital distance. One could fit a trend line to these remaining planets and find a similar CO₂-incident flux relationship to the one proposed in the paper, all without any actual climate regulation! Observing a trend of increasing CO₂ with decreasing incident flux among just planets with temperate climates is not a test of the carbonate-silicate cycle climate regulation hypothesis, as climate regulation is not required to produce this trend.

The paper needs to more clearly lay out their hypothesis for carbonate-silicate cycle climate regulation, and how it would be tested, in light of this issue. A better test might be the percentage of planets that end up with habitable versus non-habitable climates; if there is active climate regulation we'd expect a much higher proportion of planets to end up in the temperate climate range than if there is no climate regulation, and atmospheric CO₂ levels are totally random from planet to planet. The authors could then estimate how many planets would need to be observed to test these expectations. Another possibility is just considering all planets, even if they result in non-habitable climates, in both the models and observations. A trend of increasing CO₂ with decreasing incident flux among all planets observed in the habitable zone would serve as a test of whether climate regulation occurs. If we stick with just planets that have temperate climates, the CO₂-incident flux relationship for the random CO₂ case and a case with active climate regulation might have different functional forms or different slopes that could be potentially distinguished with observations. The authors could then determine how many planets we'd need to observe to distinguish between these different CO₂-incident flux relationships.

This is an excellent point and something we should have considered in previous versions. Upon inspection, the S - $p\text{CO}_2$ trendline from random $p\text{CO}_2$ (between uninhabitable upper and lower surface temperature bounds) is indistinguishable from that of the weathering mediated trendline. Rather than test the linear relationship between S and $\log(p\text{CO}_2)$, we have modified our manuscript to consider the two-dimensional relationship between S and $\log(p\text{CO}_2)$, similar to the suggested method above. This new analysis has replaced section 2.3, which further outlines how future observations might test the HZ hypothesis from our predictions. Figure 3 has also been updated to reflect our new analysis.

See line 232 (the start of section 2.3)

2. Another key point is whether the models cover the full uncertainty range expected for exoplanets, or even just Earth-like exoplanets. The new discussion in the methods section and new figures 5 & 6 are really helpful here, in showing what factors drive uncertainty in the models. I appreciate that the authors are more careful now to state that they are assuming only Earth-like planets, and real variability could be greater. But I think the paper is still overconfident that they are even capturing the full variability for Earth-like planets. I don't think the authors need to modify their model; the approach taken is a reasonable first order estimate. But the text is far too dismissive of additional uncertainty on top of what they modeled, even for just "Earth-like" planets. Stating their conclusions in this way could then be potentially misleading to readers, especially those not as familiar with how uncertain our understanding of the carbonate-silicate cycle really is.

The main argument is that the model, primarily through variations in the outgassing rate, biological mediation of weathering, and land area, produces variations in CO₂ larger than observed during periods of major climate change in Earth's past. However it matters what factors actually lead to these climate swings Earth has experienced, as different climate swings have different proposed mechanisms: changes in "weatherability" due uplift and erosion (Kump & Arthur, 1997); changes in lithology, like exposing large areas of highly reactive mafic crust (McDonald et al 2019); changes in outgassing rate, and so on. Changes in lithology or erosion rate are not included in the modeling, but either factor can lead to significant changes in weathering rates on their own. The authors argue their land fraction and biological mediation factors basically cover the range of possible changes in "weatherability." However, lumping independent factors like erosion rate or lithology into one parameter means missing possible edge cases, where the independent factors all align at one extreme or the other, unless the catch-all parameter is properly calibrated to account for this. For example, if the authors included a weatherability factor, capturing variations in erosion rate due to uplift events, in addition to f_{land} and f_{bio} , there would be cases where all three of these parameters end up at their extrema, producing a greater spread in CO₂ than the current model predicts. Just because the model can produce large CO₂ variations doesn't mean it is capturing the full uncertainty range and possible edge cases that could arise. Given that we lack a complete mechanistic understanding of how the carbonate-silicate cycle operates (the governing equations of the model this paper uses could be missing significant factors, even beyond the ideas of "weatherability" I bring up here), it is really not justifiable to say that the model is capturing the full uncertainty range in climate states, even for only "Earth-like" planets.

To clarify that uncertainties exist in current understanding of the carbonate-silicate weathering cycle, we have modified the first paragraph of section 4.4. It emphasizes that our model is an approximation based on the Earth and that the precise nature of the weathering feedback is uncertain.

Line 522 – "...weathering parameterization to reasonably approximate Earth-like planets in the HZ. However, uncertainties in how the carbonate-silicate weathering cycle regulates climate on Earth persist [19], so the predicted variations in pCO₂ in our model may not capture the true variability of pCO₂ in the HZ. Below, we show that our broad parameterization of the carbonate-silicate weathering cycle may encompass the plausible range of pCO₂ for the Earth through time, but improved understanding the carbonate-silicate weathering cycle may be necessary to know if such variations are indeed representative of the Earth through time and applicable to Earth-like planets generally."

3. Related to point 2, it is surprising that Earth sits at the lower bound of atmospheric CO₂, for the present solar flux, in Figure 1. How does sampling across all the different model parameters used not result in some cases with lower atmospheric CO₂ than Earth? The model predicts that the typical planet at Earth's incident flux would have ~100 times the atmospheric CO₂. Why is this?

This is due to our choice for parameter ranges. We consider the Earth through time to constrain our model parameters, which means the modern Earth sits at the high end of those ranges for biological weathering rate and continental land fraction, and at the low end for outgassing – the parameters that drive pCO₂ in our model. We added a new paragraph to the methods section to explain this and stressed again that our model is only applicable to planets like the Earth through time.

Line 442 – "The parameter ranges shown in Table 1 represent the uncertainty of the carbonate-silicate weathering cycle on the Earth through time [19]. Implicit in our assumed parameter ranges is that continental land fraction, f_{land} , and biological weathering fraction, f_{bio} , have increased from 0 when the Earth formed to 1 on the modern Earth. Similarly, the relative internal heat, Q , is assumed to be large when the Earth is young and unity for the modern Earth. Therefore, on the modern Earth, where $f_{land} =$

1, $f_{\text{bio}} = 1$, and $Q = 1$, the weathering rate is maximized and outgassing rate is relatively small (see Section 4.4 for a discussion on the importance of these three parameters in our model). This is seen in Figure 1, where the modern Earth appears near the lower bound for predicted $p\text{CO}_2$ in the HZ. If the continents on an exoplanet were more easily weathered or outgassing much lower than on the modern Earth, such exoplanets could have $p\text{CO}_2$ values well below the modern Earth value shown in Figure 1. We do not consider such exoplanets in this model, so the results presented here are only applicable to planets similar to the Earth through time.”

4. At the end of section 4.3, the authors state that it can take ~ 1 Gyr for the models to reach a steady-state. This is surprising, as usually the weathering rate is expected to adjust and balance outgassing more rapidly, on the order of millions of years. So why do some models take so long to reach a steady-state? This is an important issue, because the model results in Figure 1 assume a steady-state in the carbonate-silicate cycle. Rates of volcanic outgassing change over hundreds of millions of years to Gyr timescales (e.g. Tajika & Matsui 1992, Foley & Smye 2018, Dorn et al 2018), so if similar timescales are needed for steady-state in the carbonate-silicate cycle to be established, then planets will be unlikely to actually be in such a steady-state. Atmospheric CO_2 levels on real planets could then be in a transient state, and be very different than the model predictions.

Most of the parameter combinations in our model converge on short (Myr) timescales but can take up to a few 10s of Myr. Our description of the 1 Gyr convergence time was in reference to our extrapolation of the rate of change for $p\text{CO}_2$ to check for steady state. We have adjusted the wording of our model description to make this clear. Under rare parameter cases, it can take more than 10s of Myr for convergence. However, these longer convergence times may be an artifact of our model, as we start each run with the modern Earth and implement a step change in the system parameters. It is perhaps unlikely for a planet to have outgassing and weathering rates change by many orders of magnitude in an instant, which leads to artificially long convergence times. To clarify our model approach, we have modified the wording.

Line 503 – “For each parameter combination in our simplified model, we start with the modern Earth then impose a step change for each model parameter. We then run the simulation for 10 Gyr or until the system reaches steady state. We consider the model to have reached steady state when extrapolation of the rate of change of $p\text{CO}_2$ for 1 Gyr changes $p\text{CO}_2$ by less than 1%. Typically, the model converges within a few Myr to a few tens of Myr. Rarely (2 of the 1200 planets simulated in this work), parameter combinations will not reach steady-state after 10 Gyr. Simulations with combinations of exceptionally high outgassing rates and low CO_2 weathering rates can enter a regime where atmospheric CO_2 builds without bound, never converging. Such model results are beyond the range of validity of our model.”

Minor comments:

Lines 73-77: This isn't stated quite right. It's not that any volcanic outgassing will lead to an increase in surface temperature and weathering rates. To cause an increase in surface temperature, and kick off the stabilizing feedbacks of the carbonate-silicate cycle as described, requires an increase in outgassing rate above the previously established steady-state between weathering and outgassing.

Wording updated; it now reads:

Line 73 – “Carbon returns to the atmosphere from outgassing. If CO_2 outgassing increases above the steady-state outgassing rate, a planet's surface temperature rises. This leads to increased rainfall and continental weathering as well as potentially warmer deep-sea temperatures and more seafloor

weathering [19, 22, 25]. Increased weathering draws down atmospheric CO₂ and stabilizes the climate over ~10⁶-year timescales on habitable, Earth-like planets [26].”

Equation 4: I don't understand how the exponent c here incorporates both silicate and carbonate weathering. Since carbonate weathering doesn't lead to net CO₂ drawdown, usually only silicate weathering is included in developing simple weathering models like the equations developed in section 2.2. Is equation 4, and subsequent equations, meant to incorporate both silicate and carbonate weathering? If so it would be more clear to write things as a sum of these two weathering equations, or at least spell this out more directly. Or if equation 4 is just the silicate weathering rate, then it would be more clear to just use α instead of defining the new parameter c .

For the derivation in 2.2, we do not consider seafloor weathering so only continental silicate weathering is relevant. As suggested, we have replaced c with the silicate weathering exponent from Table 1, α , to avoid confusion.

Line 376: “discusses” should be “discuss”

Fixed

Reviewer #2 (Remarks to the Author):

Revision of: Lehmer and co-workers “Carbonate-Silicate Cycle Predictions of Earth-like Planetary Climates and Testing the Habitable Zone Concept”

The authors have addressed the concerns and suggestions raised by the first round of review. The revised manuscript was a pleasure to read, and my comments below are aimed at clarifying the details of the model, which I think will increase the impact of this research.

Comments:

My reading of the literature is that the authors constrained their model by adopting the reasonable temperature range of 248-355 K. The obtained distribution of stable climate solutions presented in Figure 1 basically follows this, but there is also a void where no solution exists even at >248 K for an incident flux of >~0.65. I am struggling with whether the authors have a mechanism whereby such solutions are precluded. My sense is no, and I am not sure why the model cannot find any solutions there. Is it simply because the parameter ranges in Table 1 do not explore such solutions? In other words, I'd like to understand what controls the lower boundary of the expected distribution of stable climates. It maybe that I am really missing the point here, but in this case this should be clarified. I think this would be an important thing to consider, because the authors' following arguments (e.g., Fig. 3) may be affected if the solutions exit there when the wider parameter ranges are explored.

The lack of steady-state solutions in the regions mentioned above is an expected result of the carbonate-silicate weathering cycle. It exists because, as incident flux decreases, pCO₂ must increase due to the pCO₂- and temperature-dependent nature of the weathering feedback. These characteristics play an important role in our revised test of the HZ hypothesis and are now discussed in Section 2.3.

See the paragraph starting on line 258.

I would recommend this manuscript for publication, but only after the authors discussed or dealt with above remaining concern.

REVIEWERS' COMMENTS

Reviewer #1 (Remarks to the Author):

The authors have carefully addressed all of my comments from my previous review with their revision. I recommend the paper be accepted for publication.

Reviewer #2 (Remarks to the Author):

The authors have addressed my concerns. They have also done a revised analysis responding to the concerns of another reviewer as well. I think this is a nice paper that will stimulate further debate and consideration within the community. I recommend this manuscript for publication.